## PERSPECTIVE

# Heartificial intelligence: smart solutions for CHF: An A(I)MT approach

Samuel Gillman[1], Irving H. Zucker[2] (ID) and Han-Jun Wang[3] (ID)

[1]*Department of Genetics Cell Biology and Anatomy, University of Nebraska Medical Center, Omaha, NE, USA*

[2]*Department of Cellular and Integrative Physiology, University of Nebraska Medical Center, Omaha, NE, USA*

[3]*Department of Anesthesiology, University of Nebraska Medical Center, Omaha, NE, USA*

Email: hanjunwang@unmc.edu

Handling Editors: Harold Schultz & David Paterson

The peer review history is available in the Supporting Information section of this article (https://doi.org/10.1113/JP287953#support-information-section).

Myocardial infarction (MI) initiates a cascade of physiological responses that can lead to detrimental structural and functional changes in the heart. Central to these changes is the overactivation of the sympathetic nervous system, also known as sympatho-excitation. Increased sympathetic nerve activity not only exacerbates acute cardiac dysfunction, but also plays a critical role in the chronic progression to heart failure, contributing significantly to sudden cardiac death (Fukuda et al., 2015; Triposkiadis et al., 2009). Although the role of sympatho-excitation in post-MI autonomic dysfunction is well documented (Hanna et al., 2018; Wang et al., 2014), current therapeutic interventions fall short of fully addressing the complexities of the sympathetic nervous system.

A significant advancement in managing sympathetic overactivation was demonstrated by Vrabec et al. (2025), who applied axonal modulation therapy (AMT) to control the release of norepinephrine (NE) and neuropeptide Y (NPY) in the myocardium. Both neurotransmitters are strongly linked to adverse cardiac events following MI. This work potentially paves the way for a new therapeutic avenue to mitigate adverse cardiac outcomes.

A porcine model was used to evaluate the effectiveness of AMT in reducing NE and NPY release within cardiac tissue post-MI. NE and NPY levels were measured in response to cardiac stress induced by programmed pacing, specifically at the MI-affected border zone, as well as the remote, healthier left ventricular zone. For cardiac electrophysiologists, programmed pacing functions as an intra-procedural cardiac stress test that can reveal arrhythmogenic circuits otherwise hidden by sedation or anaesthesia. The potential impact of cardiac reflexes on responses triggered during pacing is often overlooked. As shown in their figure 3, NE and NPY display unique kinetic profiles in response to pacing: NE increases during pacing but returns to baseline afterward, whereas NPY remains elevated for a time after pacing ceases.

When AMT was applied to cardiac nerves during programmed stimulation, it reduced NE release in healthy hearts, specifically in the lateral and medial left ventricular regions. In MI hearts, AMT produced location-dependent effects, significantly reducing NE in the remote zone but with variable effects in the border zone. NPY levels were unaffected by AMT in healthy hearts but were reduced in the remote zone of MI hearts. The differential effects of AMT on NE and NPY following MI and programmed stimulation are intriguing, although Vrabec et al. (2025) did not explore this further in their study.

The introduction of AMT marks a pivotal step in cardiac health management. Its strength lies in its scalable, reversible and controllable approach to regulating sympathetic nerve signalling in the heart. Current pharmacological treatments often fail to comprehensively address sympatho-excitation, a primary driver of adverse events post-MI. AMT also holds translational potential for drug-resistant cardiac arrhythmias. Chemical treatments for sympathetically mediated arrhythmias typically offer short-term relief by temporarily blocking stellate ganglion (SG) activity with local anaesthetics. In cases of recurrent drug-resistant arrhythmia, patients may undergo a surgical stellate ganglionectomy (SGx), an effective but invasive procedure that carries potential off-target effects. AMT, by targeting the neural pathways responsible for sympathetic overactivity directly, enables precise modulation with potentially fewer side effects.

The differential modulation of NE and NPY in MI-affected myocardial regions highlights the value of targeted therapies that consider cardiac anatomy. Given that post-MI hearts often display a heterogeneous pattern, with varying degrees of sympathetic dysfunction, such a tailored approach is both innovative and promising. Using a large animal model to simulate human cardiac physiology adds strength to the study by Vrabec et al. (2025), underscoring the translational potential of AMT. Direct measurements of NPY and NE in both remote and border zones in response to AMT provide valuable insights into neurotransmitter dynamics under both healthy and diseased conditions. Of note, the reduced efficacy of AMT on NE release in the border zones of MI hearts may reflect structural and functional damage, such as nerve or tissue alterations.

The potential integration of AMT with wireless modulation technologies holds significant promise for the future of heart failure treatment. A wireless modulator of stellate ganglia activity would be significant in the delivery of AMT. It would allow for real-time, on-demand adjustments based on continuous monitoring of neuro-transmitter levels and haemodynamic metrics. The fusion of diagnostic and therapeutic functionalities would dovetail seamlessly with the ethos of precision medicine: personalized inter-vention strategies that optimize patient outcomes. Figure 1 summarizes current and emerging strategies targeting the stellate ganglia for treating chronic heart failure.

The Journal of Physiology

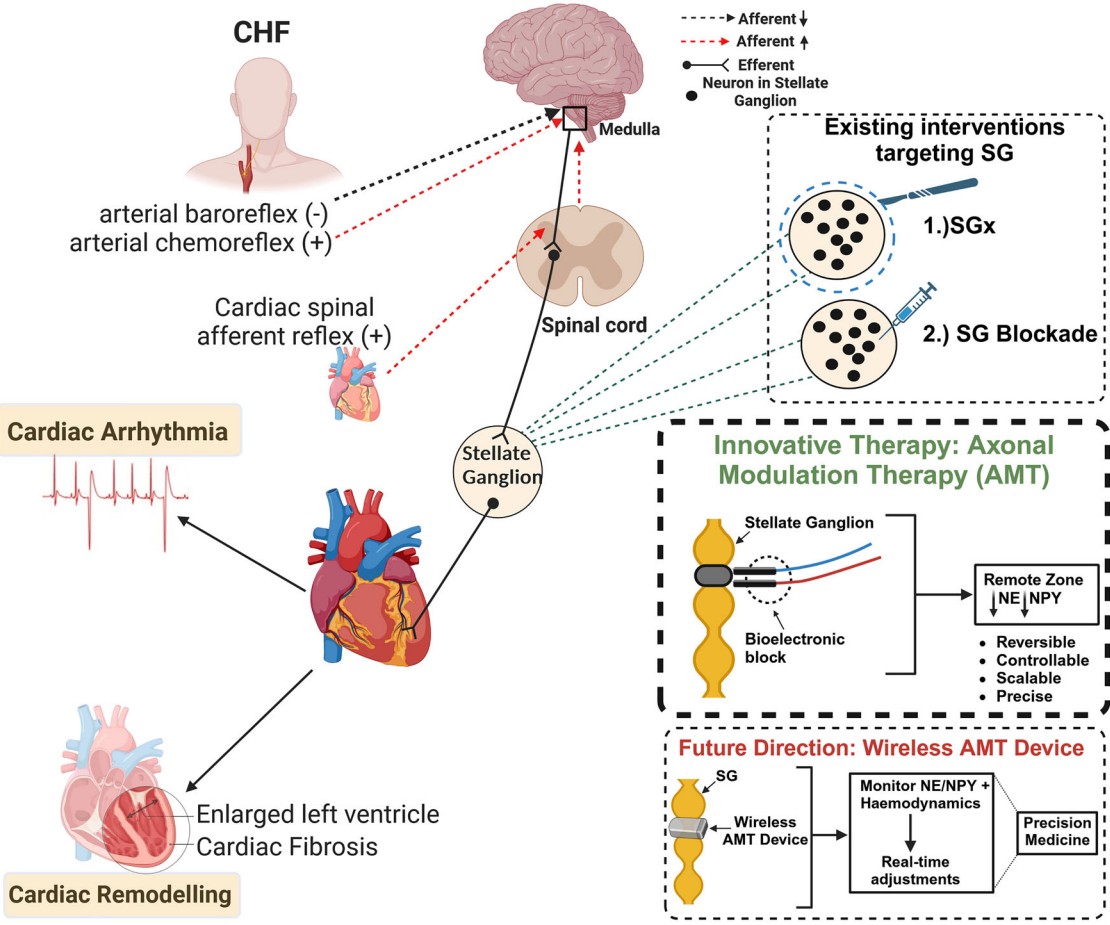

**Figure 1. Strategies for treating chronic heart failure**
A schematic illustration summarizing current and emerging strategies targeting the stellate ganglia for treating chronic heart failure.

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

## Additional information

### Competing interests

No competing interests declared.

### Author contributions

S.G., I.Z. and H.-J.W. were responsible for the conception or design of the work and drafting the work or revising it critically for important intellectual content. All authors approved the final manuacript submitted for publication. All authors agree to be accountable for all aspects of the work.

### Funding

This study was supported by NIH grants R01 HL152160, R01 HL126796, R01HL169205, R01 HL172029, R01 HL171602 and R21 HL170127.

### Keywords

autonomic dysfunction, myocardial infarction, neuromodulation, stellate ganglionectomy

### Supporting information

Additional supporting information can be found online in the Supporting Information section at the end of the HTML view of the article. Supporting information files available:

**Peer Review History**

