## [Peer Review History · The Journal of Physiology]

Heartificial Intelligence: Smart Solutions for CHF, an A(I)MT approach

Samuel Gillman, Irving H. Zucker, and Han-Jun Wang

DOI: 10.1113/JP287953

Corresponding author(s): Han-Jun Wang (hanjunwang@unmc.edu)

Review Timeline:

Submission Date:

13-Nov-2024

Accepted:

27-Nov-2024

Senior Editor: Harold Schultz

Reviewing Editor: David Paterson

Transaction Report:

Dear Dr Wang,

Re: JP-P-2024-287953 "Heartificial Intelligence: Smart Solutions for CHF, an A(I)MT approach" by Samuel Gillman, Irving H. Zucker, and Han-Jun Wang

We are pleased to tell you that your paper has been accepted for publication in The Journal of Physiology.

IMPORTANT

We do not seem to have a separate high resolution version of the figure. Can you send us this file as soon as possible, please?

You can email it to us at: jp@physoc.org

Thank you!

Yours sincerely,

Harold Schultz
Senior Editor
The Journal of Physiology

If you would like to receive our 'Research Roundup', a monthly newsletter highlighting the cutting-edge research published in The Physiological Society's family of journals (The Journal of Physiology, Experimental Physiology, Physiological Reports, The Journal of Nutritional Physiology, and The Journal of Precision Medicine: Health and Disease), please click this link, fill in your name and email address and select 'Research Roundup':

<https://www.physoc.org/journals-and-media/membernews>

- You can help your research get the attention it deserves! Check out Wiley's free Promotion Guide for best-practice recommendations for promoting your work at: www.wileyauthors.com/eoo/guide. You can learn more about Wiley Editing Services which offers professional video, design, and writing services to create shareable video abstracts, infographics, conference posters, lay summaries, and research news stories for your research at: www.wileyauthors.com/eoo/promotion.

The Corresponding Author will receive an email from Wiley with details on how to register or log-in to Wiley Authors Services where you will be able to place an order

EDITOR COMMENTS

Reviewing Editor:

Very good.

Senior Editor:

The editors thank the author for these final adjustments to their manuscript. The perspective article is now accepted for publication. Congratulations for an interesting and insightful study. Please consider the Journal of Physiology for your future studies.

REFEREE COMMENTS

Referee #1:

No additional comments